# The First Data on Harpacticoid Copepod Diversity of the Deep-Water Zone of Lake Baikal (Siberia, Russia)

Elena B. Fefilova [1,*], Tatyana Y. Sitnikova [2] and Aleksandr A. Novikov [3]

1 Institute of Biology, Komi Scientific Centre, Ural Branch of the Russian Academy of Sciences, 28 Kommunisticheskaya St., 167982 Syktyvkar, Russia

2 Limnological Institute, Siberian Branch of the Russian Academy of Sciences, 3 Ulan-Batorskaya St., 664033 Irkutsk, Russia

3 Department of Zoology, Kazan Federal University, 18 Kremlyovskaya, 420008 Kazan, Russia

* Correspondence: fefilova@ib.komisc.ru

**Abstract:** Lake Baikal (LB) is the only freshwater ancient lake on Earth where animals inhabit all bathymetric zones down to the deepest sites (approximately 1640 m). However, there is very little data on the composition of their abyssal harpacticoid fauna. On the basis of the samples collected in LB in 2010–2017 at a depth of 270–1632 m, analysis of the fauna composition and species diversity of harpacticoids in the deep-water zone is presented. Studies were conducted in all parts of the lake, including areas of a hydrothermal, oil–methane seeps, and mud volcanoes. Nineteen Baikal endemic morphological species of the genera *Bryocamptus*, *Attheyella*, and *Moraria* (*Baikalomoraria*) were found. A brief description of the taxa morphology is presented. The genus *Bryocamptus* was the richest by species number at the studied sites, and *Bryocamptus smirnovi* Borutzky was the most frequent. The most diverse (8 species) was the fauna of the Saint Petersburg methane seep. Studies have shown that the taxonomic diversity of harpacticoid copepods in the deep-water zone of LB is lower than in its littoral zone. According to two non-parametric species estimators (Chao 2 and Jackknife 1), a 1.5-fold increase of species richness of harpacticoids of the LB abyssal is expected.

**Keywords:** harpacticoids; Lake Baikal endemics; morphological species; predicted species richness; *Bryocamptus*; *Baikalomoraria*; underwater gas-hydrate-bearing structures; mud volcanoes; hydrothermal seep



## 1. Introduction

As in other ancient lakes, the fauna of Lake Baikal (LB) is characterized by the following features: a high level of endemism and a disproportionately high diversity of certain groups of invertebrates, which, under conditions of sympatric morphogenesis and speciation, form the so-called species flocks (rapidly diversifying taxonomic groups) [1]. These features are manifested in meiobenthos copepods—harpacticoids, which in terms of the number of known species (78), occupy the third place among crustaceans in the lake and account for two-thirds of the species richness of copepods; of the harpacticoid species in the lake, 85% are endemics [2,3].

One of the main trends in the evolution of benthic invertebrates of LB is habitat partitioning by depth [4]. Although the area of the abyssal zone in the lake significantly exceeds the area of the shallow zone, the deep-water harpacticoid fauna has been studied very poorly. Of all the species of Harpacticoida known for LB, most are registered at depths from several meters to several tens of meters (up to one hundred meters) [3]. Twelve of them have been constantly or occasionally registered at depths up to 200–300 m, and only one species (*Pesceus baikalensis* Borutzky, 1931), at depths up to 300–400 m. At "maximum depths" [3] (p. 480), the harpacticoid—*Bryocamptus parvus* Borutzky, 1931—has been previously found; five more unidentified and likely new representatives of the genera

*Bryocamptus* and *Moraria* have been found at depths of more than 1500 m [2,3], which makes it possible to assume the existence of a specific deep-water harpacticoid fauna in LB.

Lake Baikal appears to be the only freshwater lake which, in the deep-water zone, similar to the depths of the seas and oceans, has hydrothermal vents, cold methane and oil–methane seeps, and mud volcanoes. To date, several dozens of such formations have been discovered, mainly in the southern and middle parts of LB [5]. As in the seas and oceans, methane seeps in the lake are located in zones of tectonic activity but have a different chemical composition of mineralized fluids and hydrocarbons entering the bottom sediments and the water [5–9]. This phenomenon significantly affects the processes of relief formation in various parts of the bottom and the diversity of microscopic benthic invertebrates, as well as turbellaria, rotifers, nematodes, ostracods, cyclopoid copepods, and harpacticoid copepods [6,9,10]. In LB, animals inhabit all bathymetric zones down to the deepest sites (about 1640 m).

It is important to understand that although studies of Baikalian harpacticoid copepods have been ongoing for more than a hundred years (since 1908, when the first Baikalian species has been described), this group remains far from fully studied, even in the littoral of the lake [3]. Continuity in research is hampered by the fact that the original descriptions of most of the Baikalian endemic species are very brief and are accompanied by drawings of insufficient quality. At the same time, it is known [11] that these species are highly morphologically variable, and at least some of them may represent complexes of taxa. Using molecular methods, this assumption was confirmed, for example, for the Baikalian subendemic *Harpacticella inopinata* Sars (fam. Harpacticidae) [12]. For a complete understanding of the taxonomic diversity of harpacticoids of LB, it is, therefore, necessary to carry out their revision (starting with the revision of individual taxa) using an integrative approach (morphological and genetic) with the material obtained both from the littoral of the lake and from its deep-water zone.

The aim of our work was to obtain new data on the harpacticoid fauna of the great depths of LB, including sites of gas- and oil-bearing fluids, and to carry out the first preliminary assessment of its diversity. Our objectives included the description of the morphology of the discovered harpacticoids, their variability, the identification of taxa, and the assessment of their quantitative development in different biotopes of the abyssal zone.

## 2. Materials and Methods

### 2.1. Study Sites

Lake Baikal is located in the south of Eastern Siberia at an altitude of 455 m above sea level. The lake stretches from north to southwest for 636 km, and its width is 25–80 km. The transparency of the water in the lake is up to 40 m, and the water mineralization is predominantly approximately 100 mg $L^{-1}$. In summer, the water temperature in the bays reaches 23 °C; however, on average, the temperature in the surface layers does not exceed 8–9 °C, while in the deep layers, it is stable throughout the year at approximately 4 °C. The deep-water zone of LB is subdivided into upper abyssal (250–500 m) and low abyssal (deeper than 500 m) [13].

A brief description of sampling areas and main habitat characteristics is given according to published data [5,6,9,14–17] and the results of viewing videos received by stationary cameras installed on the staffed submersible *Mir*.

The Frolikha hydrothermal vent (or seep) is located on the slope of the northeast part of LB (Figure 1a,g and Table 1). The heat flow here arises due to the seepage of sedimentary waters through the bottom sediments. The temperature of the bottom layer in the sites of water seepage in the Frolikha hydrothermal seep is ~10 °C. It is characterized by a variety of bottom habitats; rounded pebbles and boulder covered by sponges and lying on soft sediments with rare filaments of sulfur bacteria (Figure 1e), soft oxic aleurite silts with or without sand, and bacterial mats (Figure 1a) are present.

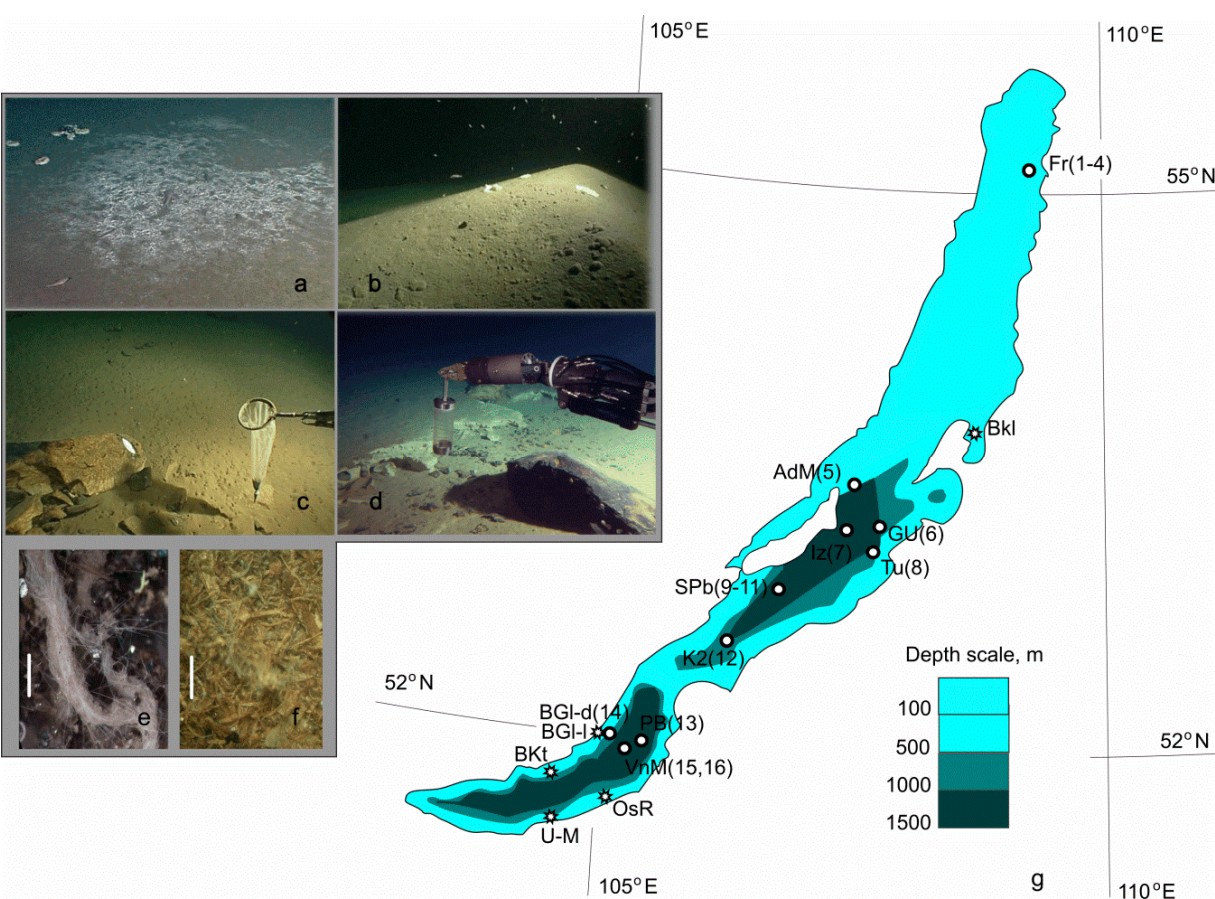

**Figure 1.** Photographs of Lake Baikal bottom (**a**–**d**) taken by underwater submersibles *Mir*, washed sediments (**e**,**f**), and map of the locations of sampling stations (**g**). (**a**)—Frolikha hydrothermal seep, bacterial mat, and sponges; (**b**)—hills of the Saint Petersburg methane seep with white flatworms on the apex and microbial mats; (**c**)—bottom near Izhimei Cape, benthic net—equipment of the *Mir*; (**d**)—Academic Ridge, gravel, boulders and exposed mudstone, benthic core—equipment of the *Mir*; (**e**)—threads of the sulfur bacteria; and (**f**)—silt and land-plant detritus. Scale (**e**,**f**): 1 cm. Circles—studied sites of deep-water zone; asterisk—studied sites of depths 10–100 m (abbreviation, see Tables 1 and 2), in brackets samples Nos.

The Saint Petersburg methane seep (Figure 2b) and the Mud Volcano Malenky are located on the lake floor (or lake bed) in the Central and Southern Baikal basins (Figure 1g) and occupy similar depth zones of ~1400 m and ~1300 m, respectively. Both seeps consist of hills formed by gas hydrates covered with a layer of soft sediments. The biotope on the Mud Volcano Malenky, from which samples were taken, is located on the slope of the hill and consists of aleurite, including iron–manganese crusts and conglomerates of diatom detritus (Table 1). The biotope at the mud volcano K-2, located in the water area of influence of the Selenga delta at the depth ~900 m (Figure 1g), is characterized by the presence of land-plant detritus in soft bottom sediments (Figure 1f). At the oil–methane seep Gorevoy Utes, the sample was taken outside the zone of oil and gas outflow; the oxic bottom sediment did not contain paraffins, detritus, or filaments of sulfur bacteria (Table 1). The lake bottom landscape at the top of Akademicheskiy Ridge is represented by rare unrounded boulders of various sizes, lying on soft bottom sediments and argillite (Figure 1b); a benthic sample was taken in the area of action of a mud volcano—there was a piece of gas hydrate at the bottom of the grapple. The bottom of the lake near Izhimei Cape was heterogeneous, soft bottom sediments are prevalent (Figure 1c) and have numerous pores and pits; white flatworms and sponges were found on scattered and different-sized boulders and gravel. Initially, this area was chosen as a reference site (non-seeps area), but it

was discovered that there is methane emission here as well [17]. Similarly, the seep area was found to be an underwater slope near the Turka settlement [5], where, in addition to soft sediments with rare filaments of sulfur bacteria, bizarrely shaped argillites were present (Table 1). The PosolBank methane seep is located on an underwater hill, gas hydrates occur in the bottom sediment, mainly deeper than 70 cm, and oxic silts dominate in the upper layer (Table 1). Methane seep near the village Bolshoe Goloustnoe is located on a slope; a characteristic feature of the biotope from which the sample was taken is the predominance of gray silt (Table 1).

**Table 1.** Information regarding investigated samples from the deep-water zone of Lake Baikal.

| No. | Site | Abbreviation | Date | Coordinates | Depth, m | Substrata |
|---|---|---|---|---|---|---|
| | | | Northern Baikal basin | | | |
| 1 | Frolikha hydrothermal seep # | Fr | 23 July 2010 | 55.517° N, 109.767° E | 409 | grv, spg, fTh |
| 2 | —"— | —"— | 23 July 2010 | —"— | 432 | y_al, slt |
| 3 | —"— | —"— | 25 July 2010 | —"— | 473 | Slt, sd |
| 4 | —"— | —"— | 25 July 2010 | —"— | 482 | br_al, fTh |
| | | | Central Baikal basin | | | |
| 5 | Akademicheskiy Ridge, mud volcano # | AdM | 8 July 2017 | 53.415° N, 107.875° E | 488 | al, cl |
| 6 | Gorevoy Utes oil–methane seep # | GU | 30 June 2016 | 53.305° N, 108.391° E | 883 | br_al |
| 7 | Near Izhimei Cape | Iz | 30 June 2017 | 53.165° N, 107.993° E | 1632 | dt, fTh, slt, cl |
| 8 | Near Turka settlement | Tu | 18 July 2010 | 52.933° N, 108.117° E | 456 | br_al, dt, fTh |
| 9 | Saint Petersburg methane seep # | SPb | 13 July 2010 | 52.883° N, 107.167° E | 1404 | al, dt, fTh, slt |
| 10 | —"— | —"— | 13 July 2010 | —"— | 1401 | jm, bl_al |
| 11 * | —"— | —"— | 15 July 2010 | —"— | 1396 | al, jm |
| 12 | Selenga region, mud volcano K-2 # | K2 | 2 July 2013 | 52.583° N, 106.767° E | 940 | br_al, dt, cl |
| | | | Southern Baikal basin | | | |
| 13 | PosolBank methane seep # | PB | 19 June 2010 | 52.038° N 105.846° E | 490–530 | al, fTh, cl |
| 14 | Bolshoe Goloustnoe methane seep # | BGl-d | 17 June 2010 | 51.983° N, 105.367° E | 270 | fTh, slt, cl |
| 15 | Mud Volcano Malenky # | VnM | 20 June 2010 | 51.922° N 105.636° E | 1368 | al, cl, cr |
| 16 | —"— | —"— | 3 July 2015 | —"— | 1393 | al, dt, cr |

#—Gas-hydrate-bearing structures; grv—gravel; spg—sponge; al—aleurite; y_al—yellow aleurite; br_al—brown aleurite; bl_al—black aleurite; slt—silt; dt—detritus; sd—sand; cl—clay; jm—microbial mats; fTh—filaments of colourless sulfur bacteria in the genus *Thioploca*; cr—iron-manganese crusts; and *—gas fluid.

**Table 2.** Information regarding samples from Lake Baikal used for comparisons of species morphology.

| Site | Abbreviation | Date | Depth, m |
|---|---|---|---|
| Near Baklaniy Island | Bkl | 17 August 2017 | 10 |
| Near Bolshoe Goloustnoe Village | BGl-l | 17 June 2010 | 93 |
| Near the River Bolshaya Osinovka mouth | OsR | 18 June 1968 | 45 |
| Utulik-Murina area | U-M | 12 June 1969 | 49 |
| Near Bolshie Koty Village | BKt | 17 October 1969 | 100 |

## 2.2. Field and Laboratory Methods

This study was performed in 2010, 2013, and 2015–2017 (Table 1) from aboard the research vessels (RV) *Koptyug* and *Vereshchagin*, as well as the staffed submersible *Mir* at several geographically and geomorphologically distinct sites in the Northern, Central, and Southern Basins of LB (Figure 1g).

Samples of bottom sediments at the depths 270–1632 m were collected by means of a 0.25 m$^2$ Ocean grab sampler (No. 14), a NIOZ-type box core (Nos 5–7, 12, 13, 15, and 16), and short benthos corers on the RV board or by the *Mir* (Nos 1–4 and 8–11) using a benthic net (Figure 1c), benthic scoop, and benthic corer (Figure 1d). Meiofauna samples were collected from apparently undisturbed surface layers of sediments with 1–5 subcorers (inner diameter of 6.5 cm and height of 10 cm). The sediment was washed through a net of 30 μm mesh size and fixed with alcohol or formalin in the field. In parallel with sampling, the depth was measured, and photos of the lake bottom were taken using stationary equipment installed on the *Mir*.

This article presents the results for harpacticoids from 16 semi-quantitative and qualitative samples from the deep-water zone (Table 1). In addition, for morphological comparison of individuals, we used a collection of Baikalian harpacticoids of the Zoological Museum of M.V. Lomonosov Moscow State University (MSU) and a collection of hydrobiological samples from the shallow-water zone of LB, provided by T.Ya. Sitnikova (Table 2).

Morphological features were examined using a Leica DM 4000 B microscope. Drawings were performed with a drawing tube attached to the microscope, with magnification of 1000×. The final versions of the drawings were created using the programs Adobe Photoshop CS3 Extendet and Xara Photo & Graphic Designer 6. Photographs of washed sediments were captured under stereo microscope MBS-10.

## 2.3. Specimens Examined

In this work, we assessed the diversity of morphological species of harpacticoid copepods. As morphological species, we considered both identified taxa and taxa that differed from them or had an unclear taxonomic status.

Harpacticoid taxa were identified using keys and descriptions [11,18–21]. For identification and taxa differentiation, we used morphological characters important for diagnosis of harpacticoid species: structure of caudal rami and anal operculum, structure of the fifth thoracopod, number of segments of thoracopods, armament by spinules of body somites, and structure of endopods of second–fourth thoracopod. Immature, severely damaged specimens were considered as *Harpacticoida* sp.

We used terminology of Huys and Boxshall [22], with some modifications. Abbreviations used in the text are as follows: A1—antennules, Exp—exopod, Enp(s)—endopod(s), Enp1–Enp 4—first to fourth endopod segments, P1–P5—first to fifth thoracopod.

## 2.4. Diversity Analysis

To estimate the predicted species richness, two non-parametric species estimators, Chao 2 ($S_{Chao2}$) and Jackknife 1 ($S_{jack1}$), were used [23].

When calculating $S_{Chao2}$ and $S_{jack1}$ indexes, *Harpacticoida* sp. were considered as one morphological species.

## 3. Results

### 3.1. Fauna Composition and Description of Collected Morphological Species

In total, we found 19 morphological species of harpacticoids (Table 3) belonging to three genera of the Canthocamptidae family: *Bryocamptus*, *Attheyella*, *Moraria*. The number of species per site ranged from one to eight. The most diverse harpacticoid fauna was found at the Saint Petersburg methane seep. The most common in the deep-water zone were *B. smirnovi* and *A. baikalensis*, which were found both in areas of gas-hydrate-bearing structures and in zones conditionally free from their influence; both species were found in the upper and low abyssal zones. Eleven morphological species were found in the seep

areas of the lower abyssal, at least several of which deserve description as species new to science. Four species were found to be common to the upper and lower abyssal zones.

The largest number of species was recorded on habitats with the presence of microbial mats. More than half of the examined species belonged to three subgenera of the genus *Bryocamptus*. The species affiliation of most of the collected harpacticoids could not be determined, for several reasons. First, in some samples, harpacticoids were only individual specimens and/or represented by specimens of the same sex, while reliable identification of species, especially *Bryocamptus* and *Moraria*, requires consideration of the characteristics of both males and females. Second, in most cases, despite the general similarity of the collected specimens to an already described Baikalian species, deviations from the original description were observed in one or more characteristics significant for the diagnostics of species of Canthocamptidae. Below, we provide brief descriptions of the examined morphological species.

**Table 3.** Taxa structure of harpacticoids in the studied sites of the deep-water zone of Lake Baikal.

| Taxa | Site (See Table 1) | | | | | | | | | |
|---|---|---|---|---|---|---|---|---|---|---|
| | Fr | AdM | GU | Iz | Tu | SPb | K2 | PB | BGl-D | VnM |
| *Bryocamptus* (*Bryocamptus*) cf. *abyssicola* (Borutzky, Okuneva, 1972) | | + | | | | | | | + | |
| *B.* (*B.*) cf. *sinuatus* (Borutzky, Okuneva, 1972) | | | | | | + | | | | |
| *Bryocamptus* sp. 1 | + | | | | | | | | | |
| *Bryocamptus* sp. 2 | | | | | + | | | | | |
| *Bryocamptus* sp. 3 | | | | | | + | | | | |
| *Bryocamptus* sp. 4 | | | | | | | | | | + |
| *Bryocamptus* sp. 5 | + | | | | | | | | | |
| *Bryocamptus* (*Rheocamptus*) sp. 6 | + | | | | | | | | | |
| *Bryocamptus* sp. 7 | | | | | | | + | | | |
| *B.* (*Echinocamptus*) *smirnovi* (Borutzky, 1931) | | | | | + | + | | + | | |
| *B.* (*E.*) cf. *parvus* (Borutzky, 1931) | | | | | | | | | | + |
| *Bryocamptus* sp. 8 | | | + | | | + | | | | |
| *Attheyella* (*Ryloviella*) *baikalensis* Borutzky, 1931 | + | | | | + | | | + | | |
| *Moraria* (*Baikalomoraria*) *spinulosa* Borutzky, Okuneva, 1972 | | | | | | | | | | + |
| *M.* (*B.*) *longicauda* Borutzky, 1952 | | | | | | + | | | | |
| *M.* (*B.*) cf. *sinuata* Borutzky, 1952 | | | + | | | | | | | |
| *Moraria* sp. 1 | | | | | + | + | | | | |
| *Moraria* sp. 2 | | | | | | + | | | | |
| *Moraria* sp. 3 | | | | | | + | | | | |
| Harpacticoida sp. | + | + | | + | | + | | | | + |

Order Harpacticoida Sars, 1903.
Family Canthocamptidae Sars, 1906.
Genus *Bryocamptus* Chappuis, 1929.
Subgenus *Bryocamptus* Chappuis, 1929.

*Bryocamptus* (*Bryocamptus*) cf. *abyssicola* (Borutzky, Okuneva, 1972) (Figure 2a–e,g).
*Material*: 1♀ and 2 ♂♂ from Bolshoe Goloustnoe methane seep (No 14), 1 ♂ from Akademicheskiy Ridge, mud volcano (No. 5).
*Remarks on morphology and taxonomy*. The found specimens are similar to *B. abyssicola* regarding the structure of caudal rami and anal operculum (Figure 2a,b), P5 of female (Figure 2c) and male (Figure 2g), and Enps of P3–P4 of female and male (Figure 2d,e). The outer spine of the male P4 Enp 2 is modified into a thick curved outgrowth, as in

*B. abyssicola* (Figure 2e). The specimens differed from the description of *B. abyssicola* by the presence on the anal somite of two-three long (rather than short) ventral spinules at the base of the caudal rami (Figure 2b). In addition, the apophysis of the male P3 is bilaterally serrated at the end (in *B. abyssicola* it is smooth) (Figure 2d).

*Bryocamptus* (*Bryocamptus*) cf. *sinuatus* (Borutzky, Okuneva, 1972) (Figure 3h–j).
*Material*: 4 ♀♀ from the Saint Petersburg methane seep (No. 10, 11).
*Remarks on morphology and taxonomy*. The specimens are similar to *B. sinuatus* in the structure of caudal rami and anal operculum (Figure 2h), P5 (Figure 3i), and Enps of P2–P4, but female from the sample No. 11 has P2 Enp1 and P3 Enp1 without inner setae.

*Bryocamptus* (*Bryocamptus*) sp. 1 (Figure 2f,h).
*Material*: 1 ♀ from the Frolikha hydrothermal seep (No. 3).
*Remarks on morphology and taxonomy*. The female is related in structure of caudal rami (Figure 2f), P1–P4, and P5 (Figure 2h) to *B. abyssicola* and *B. incertus* (Borutzky, 1931). These species differ well from each other in the structure of Enps of P2–P4 of males, as well as in the length of the ventral spinules of the anal somite at the base of the caudal rami. According to the latter character and the ratio of the lengths of the setae on P5 Exp (Figure 2h), this specimen differs from *B.* cf. *abyssicola* described above (Figure 2b).

*Bryocamptus* (*Bryocamptus*) sp. 2.
*Material*: 1 ♀ from the site near Turka settlement (No. 8).
*Remarks on morphology and taxonomy*. Similar to *B. incertus* in P3 armament, shape of caudal rami, and the presence of long ventral spinules on anal somite. It differs from *B. incertus* primarily by having a seven-segmented A1 (eight-segmented in *B. incertus*).

*Bryocamptus* (*Bryocamptus*) sp. 3.
*Material*: 1 ♂ from the Saint Petersburg methane seep (No. 9).
*Remarks on morphology and taxonomy*. The male differs from other *Bryocamptus* examined investigated by us in the elongated caudal rami (length/width ratio 1.5), as well as in the remarkable Enp 3 of P3 with long, loop-like seta and the ratio of setae on the baseoendopodal lobe of P5, of which the inner one is approximately two times longer than the outer one.

*Bryocamptus* (*Bryocamptus*) sp. 4 (Figure 2m–o).
*Material*: 1 ♀ from Mud Volcano Malenky (No. 16).
*Remarks on morphology and taxonomy*. The studied individual is similar to *B. longicaudatus* (Borutzky, Okuneva, 1972) in the structure of the caudal rami and armament of the posterior edges of the abdominal somites (Figure 2m) and the structure of Enp of P4 (Figure 2o). However, it has slight differences in the structure of P5 (Figure 2n) and the number of long ventral spinules on anal somite at the base of the caudal rami: *B. longicaudatus* has two spinules rather than three.

*Bryocamptus* (*Bryocamptus*) sp. 5 (Figure 4a–d).
*Material*: 1 ♀ from the Frolikha hydrothermal seep (No. 3).
*Remarks on morphology and taxonomy*. The structure of caudal rami (Figure 4a,b) and baseoendopodal lobe of P5 female from the Frolikha hydrothermal seep is similar to *B. abyssicola*. It differs from this species in the presence of setae on P3-P4 Enp 1 (Figure 4c, d) and a large number of setae/spines on P2–P4 Enp 2 (5, 6, 5, respectively).

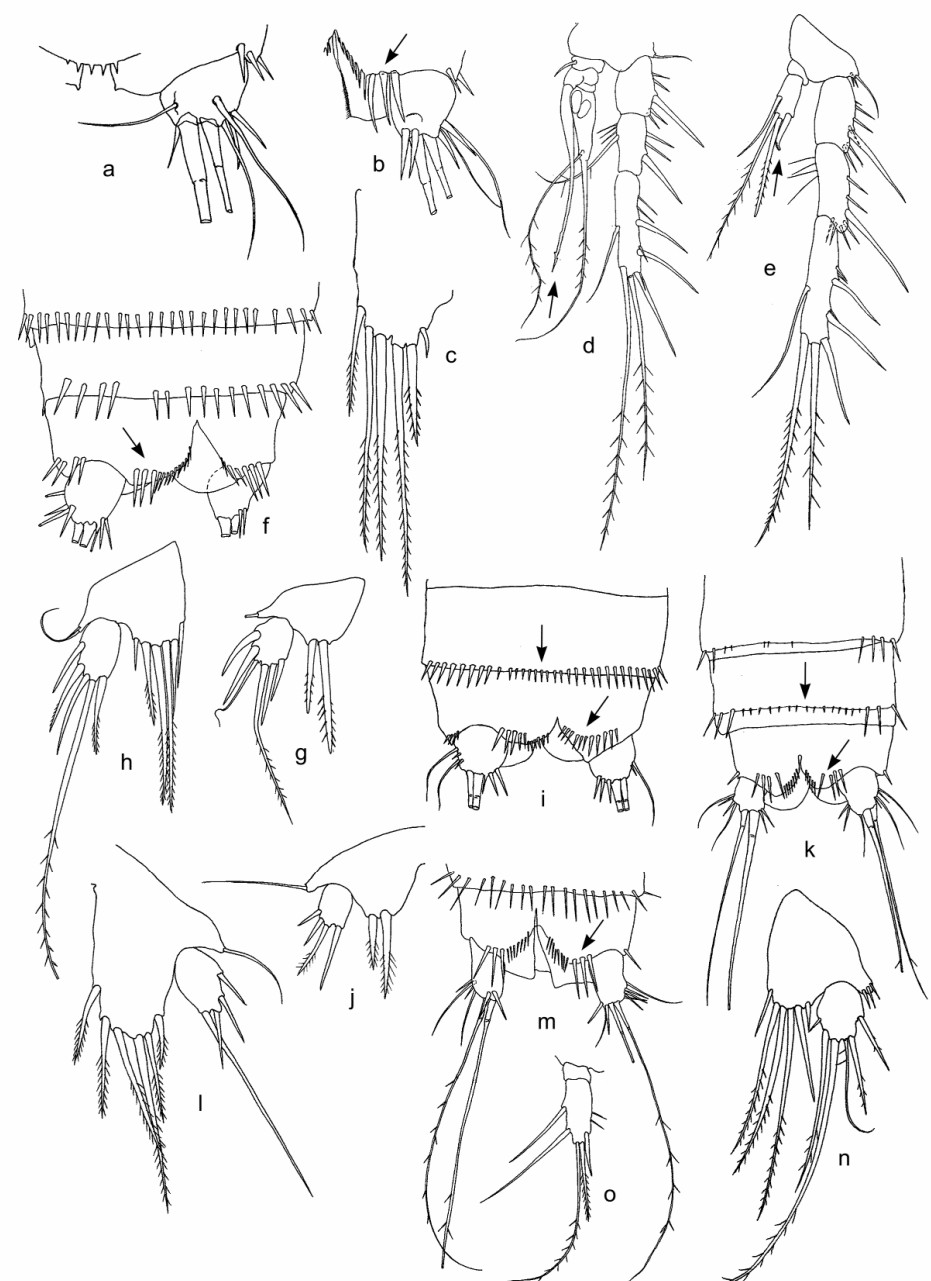

**Figure 2.** Features of females (**a**–**c**) and males (**d**,**e**,**g**) of *Bryocamptus* cf. *abyssicola* (**a**–**e**,**g**), female of *B.* sp. 1 (**f**,**h**), male of *B. smirnovi* (**i**,**j**), females of *B.* cf. *parvus* (**k**,**l**), and *B.* sp. 4 (**m**–**o**) from Lake Baikal sites: BGl-d (**a**–**c**), AdM (**e**,**g**), Fr (**f**,**h**), U-M (**i**,**j**), and VnM (**m**–**o**). (**a**)—caudal rami, dorsal view; (**b**)—caudal ramus, dorsal view; (**f**,**i**,**k**,**m**)—furca, ventral view; (**c**)—baseoendopodal lobe of P5; (**d**)—P3; (**e**)—P4; (**h**,**g**,**j**,**l**,**n**)—P5; and (**o**)—P4 endopod. Abbreviations in Tables 1 and 2. Some features are shown by arrows.

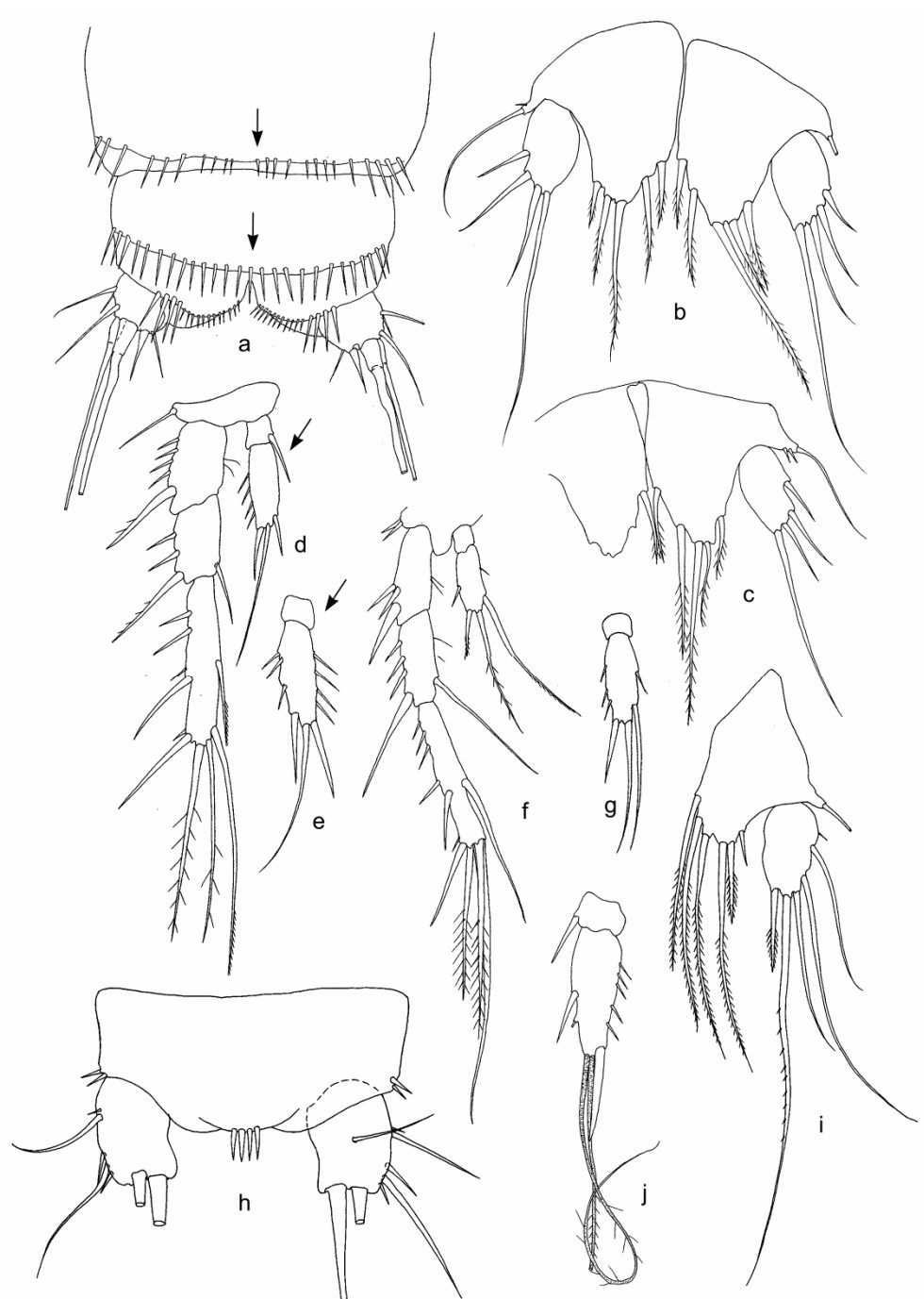

**Figure 3.** Features of females of *Bryocamptus* sp. 8 (**a–g**) and *B. sinuatus* (**h–j**) from Lake Baikal sites: Iz (**a,c,d,f**) and SPb (**b,e,g,h–j**). (**a**)—caudal rami, ventral view; (**b,c,i**)—P5; (**d**)—P3; (**e,j**)—P3 endopod; (**f**)—P4; (**g**)—P4 endopod; and (**h**)—caudal rami, dorsal view. Abbreviations in Table 1. Some features are shown by arrows.

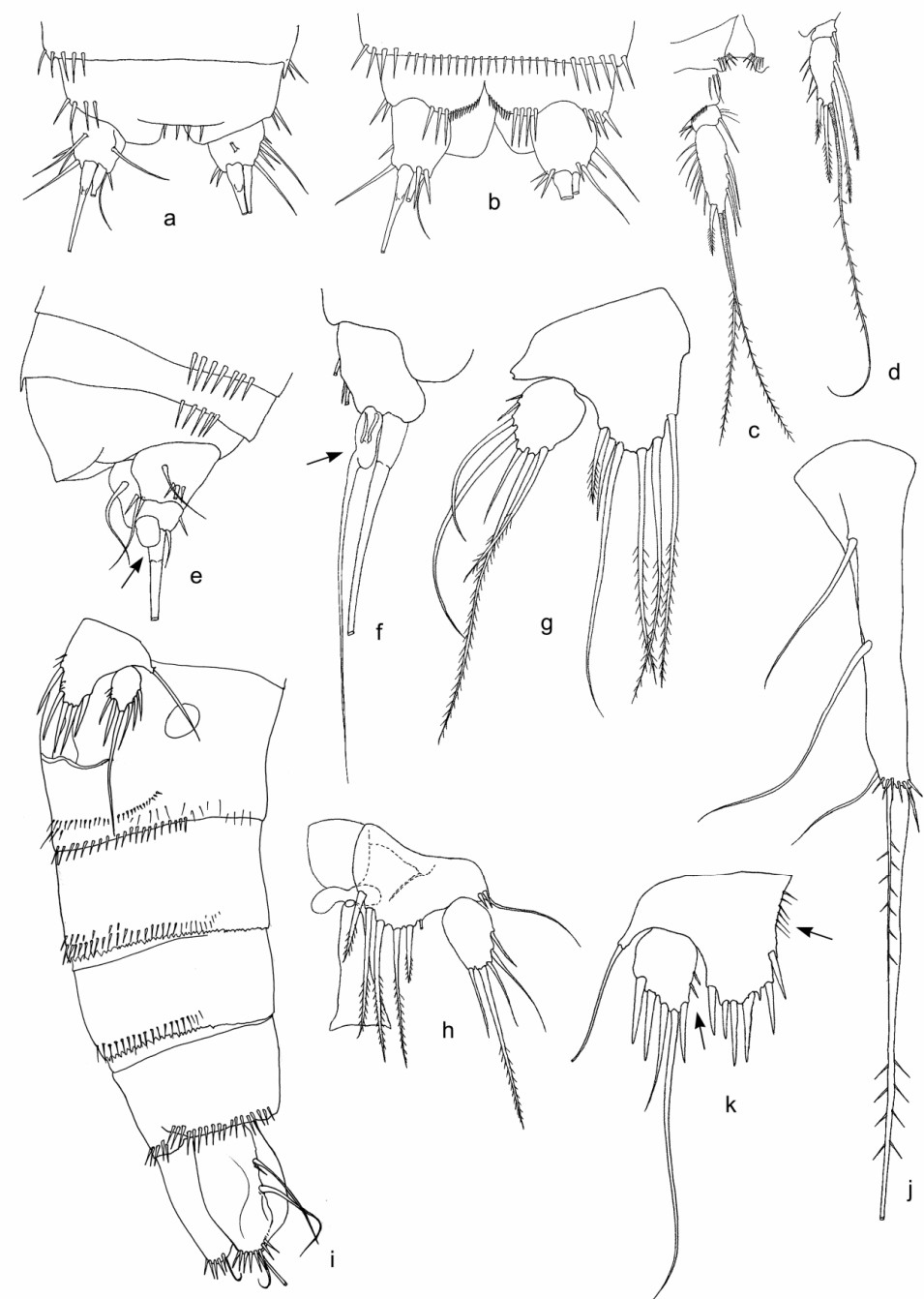

**Figure 4.** Features of females of *Bryocamptus* sp. 5 (**a–d**), *B.* sp. 6 (**e–g**), *B.* sp. 7 (**h**), *Moraria* sp. 2 (**i**), and *M.* cf. *sinuata* (**j,k**) from Lake Baikal sites: Fr (**a–g**), K2 (**h**), SPb (**i**), and GU (**j,k**). (**a**)—furca, dorsal view; (**b**)—furca, ventral view; (**c**)—P3 endopod; (**d**)—P4 endopod; (**e**)—anal somite, dorso–ventral view; (**f**)—caudal ramus, ventral view; (**g,h,k**)—P5; (**i**)—abdomen, dorso–ventral view; and (**j**)—caudal ramus, dorso–ventral view. Abbreviations in Table 1. Some features are shown by arrows.

Subgenus *Rheocamptus* Borutzky, 1952.

*Bryocamptus* (*Rheocamptus*) sp. 6 (Figure 4e–g).
*Material*: 45 ♀♀ and 3 ♂♂ from the Frolikha hydrothermal seep (Nos 1, 2).
*Remarks on morphology and taxonomy*. Compared with other representatives of the subgenus *Rheocamptus* from LB, *Bryocamptus* sp. 6 differs primarily in the structure of caudal rami (Figure 4e,f): all individuals have a laterally flattened bulb on the outer apical

seta. The P5 of the female (Figure 4g) and the male also have a characteristic structure.

*Bryocamptus* (*Rheocamptus*) sp. 7 (Figure 4h).
*Material*: 1 ♀ from the Selenga region, mud volcano K-2 (No. 12).
*Remarks on morphology and taxonomy*. The specimen is similar to *B. abyssicola* in the structure of P5 (Figure 4h) but has two-segmented Enp of P1.

Subgenus *Echinocamptus* Borutzky, 1952.

*Bryocamptus* (*Echinocamptus*) *smirnovi* (Borutzky, 1931) (Figure 2i,j).
*Material*: 1 ♀ from the PosolBank methane seep (No. 13), 43 ♀♀ and 17 ♂♂ from the Saint Petersburg methane seep (Nos 9 and 11), 1 ♀ and 1 ♂ from the site near Turka settlment (No. 8), and 5 ♀♀ and 5 ♂♂ from the Utulik-Murina area.
*Remarks on morphology and taxonomy*. The species is characterized by seven-segmented A1, three-segmented Exp and Enp of P1, smooth anal operculum, short caudal rami (Figure 2i), and four setae on female and male Exp of P5 (Figure 2j). Apparently, the species is highly variable. In the sample from the Utulik-Murina area, we found a female with one P1 branch with a two-segmented Enp and another with a three-segmented Enp.

*Bryocamptus* (*Echinocamptus*) cf. *parvus* (Borutzky, 1931) (Figure 2k,l).
*Material*: 1 ♀ from Mud Volcano Malenky (No. 16) and 3 ♀♀ from the site near Bolshie Koty Village.
*Remarks on morphology and taxonomy*. *B. parvus* is similar to *B. smirnovi* in caudal rami (Figure 2k), P2–P4, and P5 (Figure 2l). Differs from the latter mainly in the two-segmented Enp of P1. One female from the Volcano Malenky differed from the females from the Bolshie Koty Village area in the armament of the abdominal somites (Figure 2k): the row of spinules above the posterior margins of these segments was discontinuous on the ventral side; in the dorsal part of the somites, the spinules were noticeably rarer and smaller than on the lateral sides.

*Bryocamptus* (*Echinocamptus*) sp. 8 (Figure 3a–g).
*Material*: 5 ♀♀ from the site near Izhimei Cape (No. 7), 1 ♀ from the Saint Petersburg methane seep (No. 9).
*Remarks on morphology and taxonomy*. Individuals similar to *B. parvus* in caudal rami structure (Figure 3a) and to *B. smirnovi* in structure P1-P4 (Figure 3d–g) differed from these two species in P5, namely, the presence of five (not six) setae on the baseoendopodal lobe of female P5 (Figure 3b,c). Individuals from the site near Izhimei Cape differed from the specimen from the Saint Petersburg methane seep in the presence of seta on P4 Exp-1 (Figure 3d,e).

Genus *Attheyella* Brady, 1880.
Subgenus *Ryloviella* Borutzky, 1931.

*Attheyella* (*Ryloviella*) *baikalensis* (Borutzky, 1931).
*Material*: 1 ♀ from the Frolikha hydrothermal seep (No. 4), 1.♀ from the site near Turka settlment (No. 8), and 1♀ from the PosolBank methane seep (No. 13).
*Remarks on morphology and taxonomy*. The specimens examined belonged to the species *A. baikalensis* in all respects; they differed well from other Baikalian representatives of the genus *Attheyella* in the structure of the caudal rami (in *A. baikalensis*, they are elongated and covered with rows of small spinules), strong serrated posterior edges of the somites, and in the structure of P2–P5.

Genus *Moraria* Scott T. & Scott A., 1893
Subgenus *Baikalomoraria* Borutzky, 1931

*Moraria* (*Baikalomoraria*) *spinulosa* Borutzky, Okuneva, 1972 (Figure 5m–r).

*Material*: 1 ♀ and 1 ♂ from Mud Volcano Malenky (No. 15) and 2 ♀♀ from the site near the River Bolshaya Osinovka mouth.

*Remarks on morphology and taxonomy*. The morphology of the female from the Volcano Malenky almost completely matched the females (paratypes) from the site near the River Bolshaya Osinovka mouth, in particular, in the structure of the caudal rami (Figure 5m) and P5 (Figure 5o). Minor differences were observed in the structure of the caudal rami ornamentation (Figure 5m,n) and the thickness of small spinules at the inner margins of Exp and Enp of P5 (Figure 5o,p). The male was characterized by a structure of the caudal rami similar to the female, and the structure of P4 Enp (Figure 5q) and P5 (Figure 5r) were identical to those described for *M. spinulosa*.

*Moraria* (*Baikalomoraria*) *longicauda* Borutzky, 1952 (Figure 5a,b,d,i–k).

*Material*: 1 ♂ from the Saint Petersburg methane seep (No. 10) and 3 ♀♀ and 1 ♂ from the site near Baklaniy Island.

*Remarks on morphology and taxonomy*. The collected males and females were characterized by long and thin caudal rami (1.7 times longer than the anal segment) (Figure 5a,b) and particular structure of Enps of P2–P4 (Figure 5f–i). Males Exp of P5 had 5 spinules (Figure 4k). Females from the site near Baklaniy Island had 5 setae on Exp of P5, and margins of P5 were smooth (Figure 5d).

*Moraria* (*Baikalomoraria*) cf. *sinuata* Borutzky, 1952 (Figure 4j,k and Figure 5e).

*Material*: 2 ♀♀ and 1 ♂ from the Gorevoy Utes oil–methane seep (No. 6) and 1 ♀ from the site near of Bolshoe Goloustnoe Village.

*Remarks on morphology and taxonomy*. Females were similar to *M. longicauda* and *M. sinuata* in the caudal rami structure (Figure 4i) but differed from *M. longicauda* in the structure of P5 (Figures 4k and 5e).

*Moraria* (*Baikalomoraria*) sp. 1 (Figure 5c).

*Material*: 1 ♀ from the Saint Petersburg methane seep (No. 9) and 1 ♀ from the site near Turka settlment (No. 8).

*Remarks on morphology and taxonomy*. Individuals are similar to *M. longicauda* and *M. sinuata*, differing from them in the P5 structure: Exp of P5 of female from the Saint Petersburg methane seep had six setae (Figure 5c) and Exp of P5 of female from Turka area had seven setae.

*Moraria* (*Baikalomoraria*) sp. 2 (Figure 4i).

*Material*: 1 ♀ from the Saint Petersburg methane seep (No. 11).

*Remarks on morphology and taxonomy*. The female is most similar to *Moraria* (*Baikalomoraria*) *laticauda* Borutzky, 1931 in the morphology of caudal rami. As in this species, its caudal rami are greatly expanded, the apical setae are strongly reduced, and they have a row of spinules at their base (Figure 4i).

*Moraria* (*Baikalomoraria*) sp. 3.

*Material*: 1 ♀ from the Saint Petersburg methane seep (No. 11).

*Remarks on morphology and taxonomy*. The individual is similar to *Moraria* sp. 2 in the shape of the caudal rami, but the middle apical seta on them is well developed, its length is 2/3 of the length of the caudal rami. This seta is thickened at the base.

The status of most of the morphological forms of harpacticoids analyzed by us, here called morphological species, remains unclear. More detailed studies with combined morphological and molecular approach will make it possible to judge whether they are new to science or represent forms of variability in already known taxa. At least some of these morphological species can be combined into groups of undoubtedly closely related taxa. For instance, *Bryocamptus* sp. 1, *Bryocamptus* sp. 2, and *B*. cf. *abyssicola* are similar to

*B. abyssicola* and *B. incertus* (*B. abyssicola*/*incertus* group), while the three representatives of subgenus *Echinocamptus* form a common group, as does *M. longicauda*, *M.* cf. *sinuata*, and *Moraria* sp. 1.

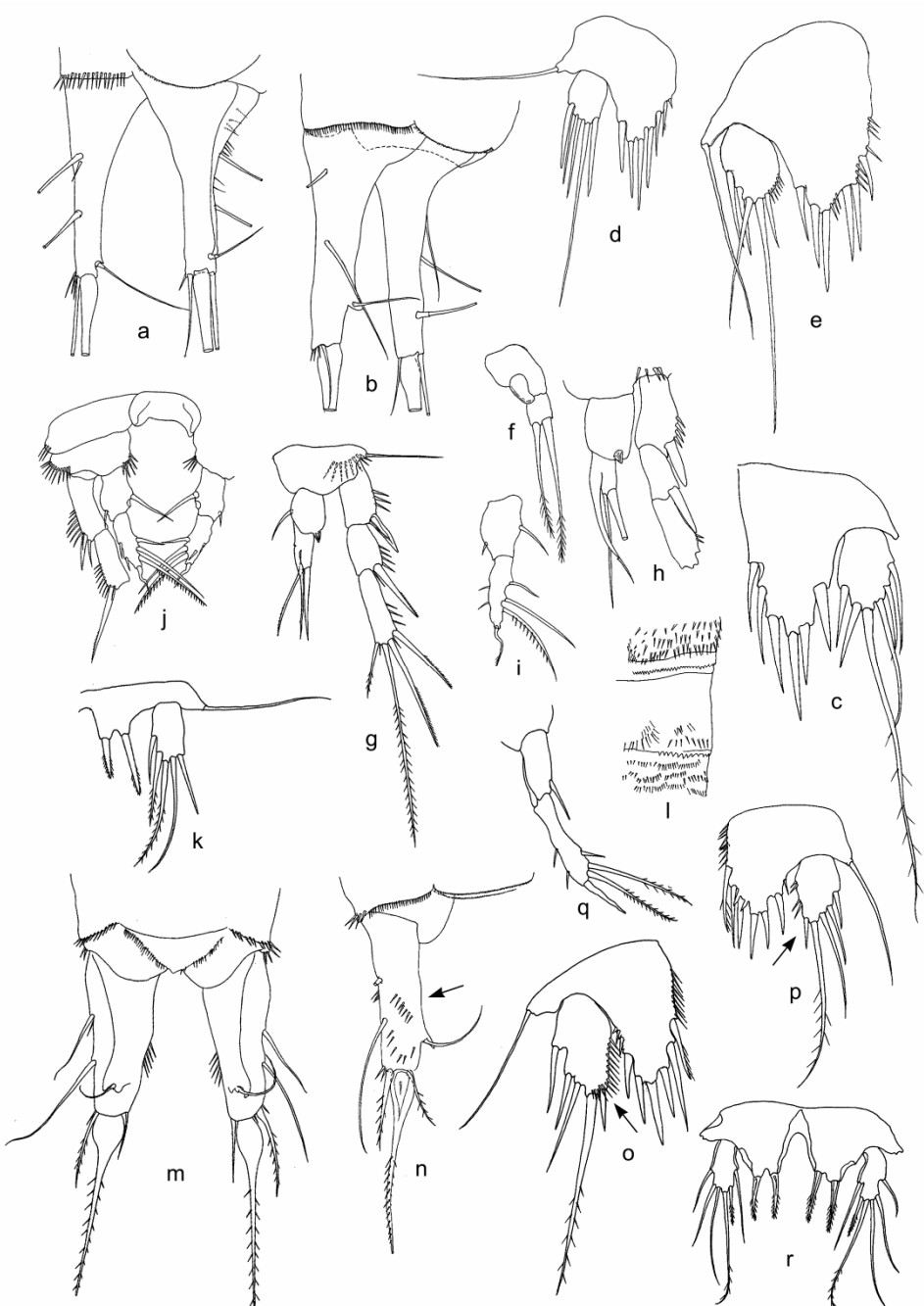

**Figure 5.** Features of females (**b,d,e**) and males (**a,f–k**) of *Moraria longicauda*, female of *M.* sp. 1 (**c**), and females (**l,m–p**) and male (**q,r**) of *M. spinulosa* from Lake Baikal sites: SPb (**a,c,h,g,i,k**), Bkl (**b,d,f,h,j**), BGl-l (**e,l**), and OsR (**m–r**). (**a,b,n**)—caudal rami, lateral view; (**c,d,e,k,o–r**)—P5; (**f**)—P2 endopod; (**g,h**)—P3; (**i,j,q**)—P4 endopods; (**l**)—arming of integument of a body somite; and (**m**)—furca, dorsal view. Abbreviations in Tables 1 and 2. Some features are shown by arrows.

### 3.2. Abundance and Diversity of Harpacticoids

Abundance of harpacticoids in some of the examined areas of the abyssal zone of LB was fairly high, although they did not belong to the absolute dominants, yielding this

role to nematodes or cyclopoids (Table 4). The highest values of harpacticoid abundance were registered at gas-hydrate-bearing structures, the Saint Petersburg methane seep and PosolBank methane seep, at microbial mats, and in the areas of concentration of biofilms of colorless sulfur bacteria of the genus *Thioploca* (Figure 1b,e), as well as in the gas-bearing fluid area (No. 11). Quantitatively, fairly abundant were harpacticoid communities of the Frolikha hydrothermal seep (Figure 1a). Representatives of the genus *Bryocamptus* dominated among the harpacticoids in these areas, while in the meiobenthos of the Frolikha seep, a large quantitative development was noted for *Bryocamptus* sp. 6, which was not found in other areas. In examined deep-water areas that were not associated with the influence of gas hydrates (areas near Cape Izhimei and settlement of Turka), the abundance of harpacticoids was not the lowest (considering that 15 specimens were found in qualitative sample No. 7). The lowest abundance of harpacticoids (a single specimen of *B.* sp. 7 was found) was registered in the area of the mud volcano K-2 on silt with land-plant detritus (Figure 1f).

Of the 20 morphological species we discovered in 16 samples, 12 species were found in only one sample, 6 in two samples. According to the calculated indexes ($S_{Chao2}$ = 28.77, $S_{jack1}$ = 31.16), the predicted species richness in the deep-water zone of Lake Baikal was 29–31 morphological species.

**Table 4.** Data regarding quantity of harpacticoids at the deep-water zone of Lake Baikal.

| No. | Site Abbreviation (Figure 1; Table 1) | Abundance (ind. m$^{-2}$) | Harpacticoid Portion in the Meiobenthos (%) | Other Groups' (Dominant) Portion in the Meiobenthos and Dominant Harpacticoid Species |
|---|---|---|---|---|
| 1 | | 1000 | 1.25 | |
| 2 | Fr | 6635 | 23.00 | *B.* sp. 6 (94% of harpacticoids abundance) |
| 3 | | 4423 | 33.00 | |
| 4 | | 2216 | 6.00 | |
| 5 | AdM | 121 | <20 | *B.* cf. *abyssicola* (100% of harpacticoids abundance) |
| 6 | GU | 909 | n.d. | *M.* cf. *sinuata* (100% of harpacticoids abundance) |
| 7 | Iz | n.d. | 11.00 | *B.* sp. 8 (~50% of harpacticoids abundance) |
| 8 | Tu | 2424 | n.d. | 4 species (25% for each species) |
| 9 | | 19,280 | 38.09 | nematodes (57%), *B.* sp. 8 (40 % of harpacticoids abundance) |
| 10 | SPb | n.d. | n.d. | *B.* cf. *sinuatus* (60% of harpacticoids abundance) |
| 11 | | 19,280 | 38.10 | *B. smirnovi* (96% of harpacticoids abundance) |
| 12 | K2 | n.d. | n.d. | n.d. |
| 13 | PB | 16,968 | 22.58 | cyclopoids (66%); *B. smirnovi*, *A. baicalensis* |
| 14 | BGl-d | 226 | 7.00 | *B.* cf. *abyssicola* (100% of harpacticoids abundance) |
| 15 | | 121 | 8.00 | *Moraria spinulosa* (67 % of harpacticoids abundance) |
| 16 | VnM | 121 | 12.50 | nematodes (38%), cyclopoids (31%); *B.* sp. 4, *B.* cf. *abyssicola* (50% for each species) |

## 4. Discussion

### 4.1. Fauna Structure

According to the data obtained, harpacticoid fauna of the deep-water zone of LB is represented by >40% of species previously found in different areas of the lake that are not associated with gas-hydrate structures [3]. However, the taxonomic diversity of harpacticoids of the deep-water zone of LB appears to be lower compared with its littoral zone. We did not find representatives of the following genera of Canthocamptidae that were earlier described for the lake [3]: *Canthocamptus*, *Maraenobiotus*, *Morariopsis*, *Epactophanes*, and the only representative of Harpacticidae, *H. inopinata*. Considering that species of the

genera *Maraenobiotus* and *Epactophanes*, which are widespread in the Palearctic, are found in Baikal exclusively at the river mouths, in the future, they are unlikely to be described among the inhabitants of the deep-water zone. This is likely also true for *Attheyella dogielli* (Rylov, 1923), *A. nordenskioldii* (Lilljeborg, 1902), *Moraria duthiei* (Scott T. & Scott A, 1896), and *M. mrazeki* Scott, 1903, which are not endemic to the lake but inhabit it. In our opinion, the list of harpacticoid taxa is likely to be expanded (according to the calculated indicators, by 1.5 times) to include representatives of the genera *Pesceus*, *Bryocamptus*, *Moraria*, and *Morariopsis*. For example, *Pesceus* sp., which we did not find, has earlier been described for benthic communities of the PosolBank methane seep [9].

Our results confirmed that another interesting feature of the LB biota, immiscibility of endemic and Siberian fauna, is strictly manifested in relation to harpacticoids: all taxa collected by us are Baikal endemic. This distinguishes the composition of harpacticoid taxocenoses in the abyssal zone from some other invertebrates present in the deep-water communities of LB. Widespread in Siberia, species of oligochaetes [24] and nematodes [10] were registered, albeit in small numbers, at the Frolikha and Gorevoy Utes seeps, respectively. The coexistence of species endemic to the lake and widespread in the Palearctic in the deep zone of LB is also characteristic of other groups of benthic invertebrates: rotifers, protozoa, tardigrades, hydroids, and acari [10].

On the other hand, the unique deep-water areas of LB may be inhabited by species that are restricted to them. For example, new-to-science oligochaete species of the Frolikha hydrothermal seep [10], as well as *Bryocamptus* sp. 6, which we detected here in high abundance, remain likely endemic to this area of the Northern Baikal basin. Findings of new-to-science harpacticoid taxa, endemic to the abyssal zone of LB or some of its areas, appear promising in light of similar discoveries for the seas and oceans [25–28]. Expansion of the faunistic list of deep-water harpacticoids will be facilitated by studying not only the composition of communities of underwater gas-hydrate-bearing structures examined in this work but also other areas of the abyssal zone of the Northern, Central, and Southern Baikal basins, where, as shown by our earlier studies, harpacticoids also occurred.

For all the species of harpacticoids described for LB that we mentioned here, the following depths were previously indicated as optimal as habitat: 5–20 m (*B. incertus*), 14–100 m (*B. sinuatus*), 9–260 m (*B. abyssicola*), 6–50 m (*B. longicaudus*), 8–15 m (*B. smirnovi*), up to 300 m (*A. baicalensis*, *M. longicauda*), 6–260 m (*M. sinuata*), and 6–100 m (*M. spinulosa*) [3,11]. Only *B. parvus* was known to be found at depths from 15 m to the maximum depth [3]. Therefore, we do not exclude the possibility that some of the morphological species collected by us are actually deep-water forms of known eurybiont taxa, and the distinctive features described by us are manifestations of the variability associated with great depth or other special conditions of the studied habitats. For example, at depths greater than 100 m, *M. longicauda* have previously been observed [11] to have caudal rami almost twice as long as those in specimens from populations in the littoral. The analysis of harpacticoid morphology presented in this work showed an extremely high level of their variability (in A1 segmentation, structure of endopods of P1–P4, P5, and structure of caudal rami). It is possible that we have observed species variability associated with the depth gradient and mosaicism of other environmental characteristics. In general, the wide distribution of transitional forms, the taxonomic status of which is difficult to determine, is an integral feature of the faunal diversity in LB [1].

### 4.2. Quantitative Aspects

With the exception of the breakers zone, harpacticoids in the Baikalian littoral zone are far from being predominant in the abundance in meiobenthic communities [3]; however, even without more detailed analysis, their abundance is clearly higher here than in the abyssal zone. For example, Okuneva [11] considered the average abundance of 2000 ind. m$^{-2}$ of *M. longicauda* in LB as low. She also noted that in the area of Bolshiye Koty Village, for example, the abundance of *A. baicalensis* was up to 34,000 ind. m$^{-2}$ (we registered this species singly). The dominant species in our samples from the Saint Petersburg methane seep *B. smirnovi* forms at a depth of 8–15 m aggregations of up to

18,000 ind. m$^{-2}$ [11], which, according to our data, is commensurate with the abundance of this species at a depth of ~1400 m.

Thus, the species identified by us can be divided into (1) widespread throughout the lake, with a high local abundance both at shallow depths and in the abyssal (*B. smirnovi*), (2) rare in the Southern Baikal basin, with abundance that decreases with depth (*M. spinulosa*), (3) widespread, with abundance that decreases with depth (*B.* cf. *abyssicola, B. parvus, A. baikalensis, M. longicauda*), and (4) unique to LB, present and abundant only in the Frolikha hydrothermal seep at depth of >400 m (*Bryocamptus* sp. 6).

Similar or opposite patterns in the distribution by depth were noted for other Baikalian groups of benthic organisms. For example, the largest diversity of nematodes (26 species, 11 genera)—compared with a reference shallow-water area, an area with methane gas bubbles, and a reference deep-water area—was recorded at a gas hydrate deep-water area of the PosolBank methane seep [9], and the quantitative indicators of the development of these organisms in the lake as a whole can vary significantly in different depth gradients and depend on the distribution of gas hydrate sites in the deep-water zone [10].

The results of our study of the harpacticoid abundance in bottom communities of the abyssal zone of LB, including sites with gas- and oil-bearing fluids, are consistent with previously obtained data. For instance, a comparative analysis of the structure of meiobenthic communities of the Frolikha, Saint Petersburg, Gorevoy Utes seeps, and Malenky volcano showed [6] that the greatest abundance of harpacticoids (up to 75,195 ind. m$^{-2}$ and 40% of meiobenthos abundance) is found in oxic silt with gas fluids and bacterial mats at the Saint Petersburg seep, while in other habitats, nematodes or ostracods were predominant. Studies of meiobenthic communities of the PosolBank methane seep have established [9] that in most quantitative samples, harpacticoids are absent or almost absent; however, according to our data, in some biotopes of this area, they developed in fairly large numbers and accounted for up to 22% of meiobenthos abundance.

All early studies on the spatial distribution of the abundance of harpacticoids at shallow- and deep-water vents and seeps were performed in marine ecosystems [7,8,29,30]. They established certain patterns, such as an increased taxonomic diversity and abundance of harpacticoids at hydrothermal vents [29] or, on the contrary, a decrease in these indicators relative to adjacent areas [8]. The factors that determine the increase in the abundance of harpacticoids in such communities include the total organic carbon concentration or/and the food amount [7,29,30], whereas the decrease in their abundance is determined by toxicity of environmental conditions (hypoxia and the presence of hydrogen sulfide) [8]. Similar to other benthic nonpredatory invertebrate [9,16], in the deep-water zone of LB, harpacticoids likely prefer the rich food resources provided by microbial mats concentrated in the so-called "oases of life" of vent and seep areas.

**Author Contributions:** Conceptualization E.B.F.; resources and data curation T.Y.S.; methodology T.Y.S., E.B.F. and A.A.N.; visualization, E.B.F. and T.Y.S.; writing, E.B.F., A.A.N. and T.Y.S.; revision, E.B.F., T.Y.S. and A.A.N. All authors have read and agreed to the published version of the manuscript.

**Funding:** The study was supported by the Russian Science Foundation (grant 22-24-00030), https://rscf.ru/project/22-24-00030/, (accessed on 20 November 2022).

**Institutional Review Board Statement:** Not applicable.

**Informed Consent Statement:** Not applicable.

**Data Availability Statement:** Not applicable.

**Acknowledgments:** The authors are grateful to the staff of the research vessels *Koptyug* and *Vereshchagin* and pilots of the *Mir* for assistance with field sampling and video materials provided. We are grateful to T.I. Zemskaya (Limnological Institute, Siberian Branch of the RAS) and the staff of the Laboratory of Hydrocarbon Microbiology of the Limnological Institute (accessed on 1 June 2010) for the sampling of meiobenthos; T.V. Naumova (Limnological Institute, Siberian Branch of the RAS) and E.S. Kochanova (University of Helsinki) for the help with processing of meiobenthos samples and providing of harpacticoids samplings; and N.M. Korovchinskiy (A.N. Severtsov Institute of Ecology

and Evolution of the RAS) and K.G. Mikhailov (Zoological Museum of MSU) for help in work with the collections of the Zoological Museum of MSU.

**Conflicts of Interest:** The authors declare no conflict of interest. The funders had no role in the design of the study; in the collection, analyses, or interpretation of data; in the writing of the manuscript, or in the decision to publish the results.

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
