# Peer review of "The First Data on Harpacticoid Copepod Diversity of the Deep-Water Zone of Lake Baikal (Siberia, Russia)"

_diversity, doi:10.3390/d15010094_

Round 1

Reviewer 1 Report

The manuscript addresses very important topic, the high, but unknown endemism of Lake Baikal harpacticoid assemblages. The MS is well written, it contains a solid set of data, the data are well reported and major finding nicely presented. I highly recommend it for the publication after minor revisison.

 Please check the minor comments in the pdf.

The font in the manuscript needs to be edited.

Author Response

Thank you for your work with our manuscript, valuable remarks and corrections. Most of them we took into account to improve the manuscript.

Comment 1: do you mean conductivity?

Response: No, we mean water mineralization – concentration of dissolved salts or ions concentration (in mg L-1).

Comment 2: I suggest to rewrite to: "A brief description of sampling areas and main habitat characteristics"...

Response: That was rewritten.

Comment 3: please add abbreviation for the sampling site in the bracket

Response: Sorry, we didn’t add abbreviations in this section. We used abbreviations only for the map (Figure 1) and in the captions to the pictures (Figures 2-5) so as not to overload them (abbreviations are presented in Tables 1 and 2). We believe that the use of abbreviations in the text (other than LB) may not be very justifiable, in most cases, double brackets would appear in the text, which would make it difficult to perceive.

Comment 4: please add information which software did you use for these calculations

Response: We didn’t use any software.

Comment 5: can you add some explanation why here so high species richness?

Response: We give some explanation in the discussion section:

…Similar to other benthic nonpredatory invertebrate [9, 16], in the deep-water zone of LB harpacticoids probably prefer the rich food resources provided by microbial mats concentrated in the so-called “oases of life” of vent and seep areas.   

We rewrote as suggested:  First, in some samples, harpacticoids were only individual specimens and/or represented by specimens of the same sex, while reliable identification of species, especially Bryocamptus and Moraria, requires taking into account the characteristics of both males and females. Second, in most cases, despite the general similarity of the collected specimens with one or another already described Baikalian species, deviations from the original description were observed in one or more characteristics significant for the diagnostics of species of Canthocamptidae. Below we provide brief descriptions of the examined morphological species.

We corrected minor mistakes too. Sorry for them, and thanks a lot!

Reviewer 2 Report

Th epaper is a quite interesting and exhaustive overview of the deep (abyssal) zone of Bajkal lakes, one of the most interesting sites on Earth. The sampling design is interesting, drawings are very accurate, and as a first analysis of these incredibly interesting environments it is a nice paper to read. It's a pity that a lot of species cannot be attributed with certainty to other previuosly described species, or considered new to Science and described. But in an area like Bajkal lake, further complicated by species flocks, genetic methods are requited in the future, so this is a good starting point.

A couple of observations to improve the paper:

(1) revision of English style and grammar; I corrected a few points, but of course a thorough revision is recommended to make the paper more fluent; I suggest that "Bajkalian" should be used as an adjective throughout the text where appropriate instead of "Bajkal" (i.e. Bajkalian species, Bajkalian abyssal zone, etc.)

(2) the pagination has to be ameliorated of course; the figure of the sampling site, very nice, is needed when reading about the area, and so on), but this is a first version

(3) maybe, as a suggestion, a complete list of all Bajkalian harpacticoids could help the readers not familiar with this peculiar fauna, and ould be added as an appendix.

Minor observations are added in the text corrections.

Author Response

Thank you for your work with our manuscript, valuable remarks and corrections. Most of them we took into account to improve the manuscript.

Comment 1: revision of English style and grammar; I corrected a few points, but of course a thorough revision is recommended to make the paper more fluent; I suggest that "Bajkalian" should be used as an adjective throughout the text where appropriate instead of "Bajkal" (i.e. Bajkalian species, Bajkalian abyssal zone, etc.)

Response: Sorry for the mistakes in English. They occured due to lack of time for the translation of complex text, which was performed by the native English speaker. We checked the English and made all your corrections. Thank you!

We add “Baikalian” instead of “Baikal” throughout the text.

Comment 2: the pagination has to be ameliorated of course; the figure of the sampling site, very nice, is needed when reading about the area, and so on), but this is a first version

Response: We are ready to provide original drawings in good quality separately.

Comment 3: maybe, as a suggestion, a complete list of all Bajkalian harpacticoids could help the readers not familiar with this peculiar fauna, and ould be added as an appendix.

Response: Sorry, we don't think it's necessary. Full list of species of Baikalian harpacticoids is published earlier (Okuneva, Evstigneeva, 2001).

Comment 4 (about formulas): These are known formulas; to report them herein  is useless and lowers readibiity; I suggest to delete them

Response: They were deleted.

Comment 5: abyssal zone (the same in the other parts of this paragraph)

Response: we add “zone” after “abyssal” or “deep-water” or “shallow-water, littoral” throughout the text.

We changed “Quantitative development” to “Quantitative aspects”,

“Similar to other benthic browsers invertebrate” to “Similar to other benthic nonpredatory invertebrate”, and corrected minor mistakes too. Sorry for them, and thanks a lot!
